# Parity transitions in the superconducting ground state of hybrid InSb–Al Coulomb islands

Jie Shen[1], Sebastian Heedt[1], Francesco Borsoi[1], Bernard van Heck [2], Sasa Gazibegovic[1,3], Roy L.M. Op het Veld[1,3], Diana Car[1,3], John A. Logan [4], Mihir Pendharkar [5], Senja J.J. Ramakers[1], Guanzhong Wang[1], Di Xu [1], Daniël Bouman [1], Attila Geresdi [1], Chris J. Palmstrøm[4,5], Erik P.A.M. Bakkers [1,3] & Leo P. Kouwenhoven[1,6]

The number of electrons in small metallic or semiconducting islands is quantised. When tunnelling is enabled via opaque barriers this number can change by an integer. In super-conductors the addition is in units of two electron charges ($2e$), reflecting that the Cooper pair condensate must have an even parity. This ground state (GS) is foundational for all superconducting qubit devices. Here, we study a hybrid superconducting–semiconducting island and find three typical GS evolutions in a parallel magnetic field: a robust $2e$-periodic even-parity GS, a transition to a $2e$-periodic odd-parity GS, and a transition from a $2e$- to a $1e$-periodic GS. The $2e$-periodic odd-parity GS persistent in gate-voltage occurs when a spin-resolved subgap state crosses zero energy. For our $1e$-periodic GSs we explicitly show the origin being a single zero-energy state gapped from the continuum, i.e., compatible with an Andreev bound states stabilized at zero energy or the presence of Majorana zero modes.

[1] QuTech and Kavli Institute of Nanoscience, Delft University of Technology, 2600 GA Delft, The Netherlands. [2] Microsoft Quantum, Microsoft Station Q, University of California Santa Barbara, Santa Barbara, CA 93106, USA. [3] Department of Applied Physics, Eindhoven University of Technology, 5600 MB Eindhoven, The Netherlands. [4] Materials Department, University of California Santa Barbara, Santa Barbara, CA 93106, USA. [5] Electrical and Computer Engineering, University of California Santa Barbara, Santa Barbara, CA 93106, USA. [6] Microsoft Station Q at Delft University of Technology, 2600 GA Delft, The Netherlands. These authors contributed equally: Jie Shen, Sebastian Heedt, Francesco Borsoi. Correspondence and requests for materials should be addressed to J.S. (email: J.Shen-1@tudelft.nl) or to L.P.K. (email: Leo.Kouwenhoven@Microsoft.com)

A superconductor can proximitize a semiconductor and open a gap in its energy spectrum. If the two materials are strongly coupled, the induced gap can be as large as the original gap in the superconductor. The two gaps respond differently to an applied magnetic field, e.g. when the Landé $g$-factors differ in the two materials. A large $g$-factor in the semiconductor can cause the induced gap to close long before the closing of the original gap. If, in addition, the semiconductor has strong spin–orbit interaction, the induced gap can re-open, signalling a transition to a topological superconducting phase[1,2]. This phase contains pairs of Majorana zero modes (MZMs) that can accommodate either zero or one fermion, and thus allows for both even-parity and odd-parity ground states (GSs)[3,4].

When a conductor has a finite size, it forms an island restricting the charge to an integer times the elementary charge, $e$[5]. The resulting Coulomb blockade effects have been widely studied in metallic and superconducting islands, the latter often referred to as Cooper pair boxes[6,7]. A major breakthrough was the demonstration of charge quantisation in units of $2e$ in aluminium (Al) islands[8–13], indicating that the even-parity superconducting GS was not poisoned by quasiparticles on the time scale of the measurement. The $2e$ quantisation could be destroyed by subjecting the Al to an external magnetic field, $B$, which causes a transition to the metallic state with $1e$ charge quantisation[10,11].

Hybrid superconducting–semiconducting islands have also shown a $2e$ charge quantisation at low $B$-fields[14–16]. These observations imply that the low-energy spectrum in the semiconductor is completely proximitized with no Andreev bound states (ABSs) at low energies. Also, for these hybrid islands a $B$-field can cause a $2e$ to $1e$ transition[15,16]. A recent breakthrough demonstrated that under particular circumstances the $1e$ quantisation is not due to the transition to the metallic state—but rather due to a topological superconducting phase[16]. These pioneering experiments used InAs as the semiconductor.

Here, we harness the large $g$-factor ($g \approx 50$) and the ballistic transport properties of InSb nanowires[17,18], and find additional $B$-field-induced transitions, including a recurrence of a $2e$ quantisation at higher $B$-fields.

## Results

**Different $B$-field GS evolutions at controllable gate configurations**. Our device (Fig. 1a) consists of a hexagonal InSb nanowire with two of its facets covered by a thin epitaxial layer of Al (see ref. [19] for materials details). Two top gates (TG) can induce adjustable tunnel barriers separating the InSb–Al island from the two normal leads. The voltage, $V_{PG}$, applied to the top plunger gate (PG), can be used to tune the charge on the island, as well as the spatial charge density profile in the semiconductor. A bias voltage, $V_b$, is applied between source (S) and drain (D), yielding a current, $I$, that is measured in a dilution refrigerator at a base temperature of ~15 mK.

Figure 1b shows the differential conductance, $dI/dV_b$, vs. $V_b$ and the voltage applied to the tunnel gate at $B = 0$. The charge is fixed in the current-blockaded Coulomb diamonds (with mostly blue colour) with a periodicity in gate voltage corresponding to a charge increment of $2e$. For $V_b > 120\,\mu V$ the periodicity is halved to $1e$, indicating the onset of single electron transport. Linecuts in the right panel show that $dI/dV_b$ can be enhanced as well as suppressed, even down to negative values (black colour). These are known features for hybrid islands and can be used to extract values for the charging energy, $E_c = e^2/2C \approx 25\,\mu eV$ and the lowest-energy subgap state, $E_0 = 50$–$90\,\mu eV$ (Supplementary Fig. 2). The Al superconducting gap, $\Delta = 220\,\mu eV$, is extracted from tunnelling spectroscopy measurements (Supplementary Fig. 3).

To understand parity transitions induced by a $B$-field, we illustrate four different scenarios in Fig. 1c–f. We sketch a slow reduction of $\Delta$ for a $B$-field, $B_\parallel$, along the nanowire axis (Fig. 1c–e) and a rapid decrease of $\Delta$ for a perpendicular field, $B_\perp$ (Fig. 1f). We consider a single lowest-energy subgap state (i.e. an ABS) with energy $E_0$, which is two-fold spin-degenerate at $B = 0$ and becomes spin-split in a $B$-field. Fig. 1c sketches the case where $E_0$ decreases very slowly with $B_\parallel$, remaining above $E_c$ such that the GS parity remains even. This translates to $2e$-periodic conductance oscillations for all $B_\parallel$-fields (Fig. 1k). Note that conductance peaks occur when the lowest-energy parabolas cross. In Fig. 1g the lowest crossings are always between even-charge parabolas. These crossings, for instance at $N_g = -1$ and $+1$, are $2e$-periodic. The odd-charge parabolas remain above these lowest-energy crossings and thus do not participate in the low-energy transport.

Figure 1d sketches a second case where $E_0$ varies more rapidly with $B_\parallel$. When $E_0$ crosses $E_c$ the odd-charge parabolas pass the lowest-energy crossings of the even parabolas, thereby adding degeneracy points between even-charge and odd-charge parabolas (Fig. 1h). This results in alternating smaller and larger peak spacings, where the smaller valleys have odd-parity for $E_0 > 0$ and even-parity for $E_0 < 0$. Note that an equal spacing with $1e$-periodicity occurs when $E_0 = 0$. At negative energies, when $E_0$ crosses $-E_c$ the $2e$-periodicity is restored, however, now with an odd-parity GS. In terms of the charge parabolas this corresponds to odd-charge parabolas being always lower in energy than the even ones (red parabolas in Fig. 1h). This case of $2e$-periodicity for an odd-parity GS has not been reported before.

In the third case we illustrate the possible consequence of strong spin–orbit interaction. The zero-energy crossing of the subgap state can now be followed by a transition to a topological phase containing MZMs rigidly fixed at $E_0 = 0$ (Fig. 1e). (Note that the re-opening of the gap is not shown in Fig. 1e since only the lowest-energy subgap state is sketched.) As $E_0$ decreases below $E_c$, the peak spacing gradually evolves from $2e$ to $1e$ with alternating even-parity and odd-parity GSs (Figs. 1i and m). It is important to note that the even/odd degeneracy of the topological phase in bulk materials is lifted here by the charging energy[3,20]. This fundamental degeneracy is however visible by comparing the lowest energies for the even and odd parity GSs, which are both zero, albeit at different gate voltages. We note that ABSs confined by a smooth potential may also give rise to a similar phenomenon as sketched in Fig. 1e[21].

Finally, in the fourth case (Fig. 1f), the superconducting gap in Al closes at its critical perpendicular magnetic field. This transition to the normal state also causes equidistant $1e$-periodic oscillations (Figs. 1j and n), which in the peak evolution is similar to the topological case. However, we show below that finite-bias spectroscopy is significantly different in these two cases.

In Fig. 2, we present exemplary data for the four cases illustrated in Fig. 1k–n. Fig. 2a–d show four panels of $dI/dV_b$ measured at zero bias as a function of $V_{PG}$ and $B$-field ($B_\parallel$ or $B_\perp$). The bottom row of panels shows representative linecuts at high $B$-fields. Fig. 2e–h show the corresponding peak spacings for even ($S_e$) and odd ($S_o$) GSs. These spacings are converted from gate voltage to energy via the gate lever arm and reflect the energy difference between even-parity and odd-parity states. In Fig. 2a the peaks are $2e$-periodic with even-parity GS up to a field of ~0.9 T. This observation reflects that up to this $B$-field our Al thin film remains superconducting without any low-energy subgap state. Above ~0.9 T the gap is significantly suppressed such that $1e$-transport sets in[11].

In Fig. 2b, the conductance peaks split into pairs around 0.11 T with alternating small and large spacings. These split peaks merge with neighbouring split peaks, leading to the recurrence of

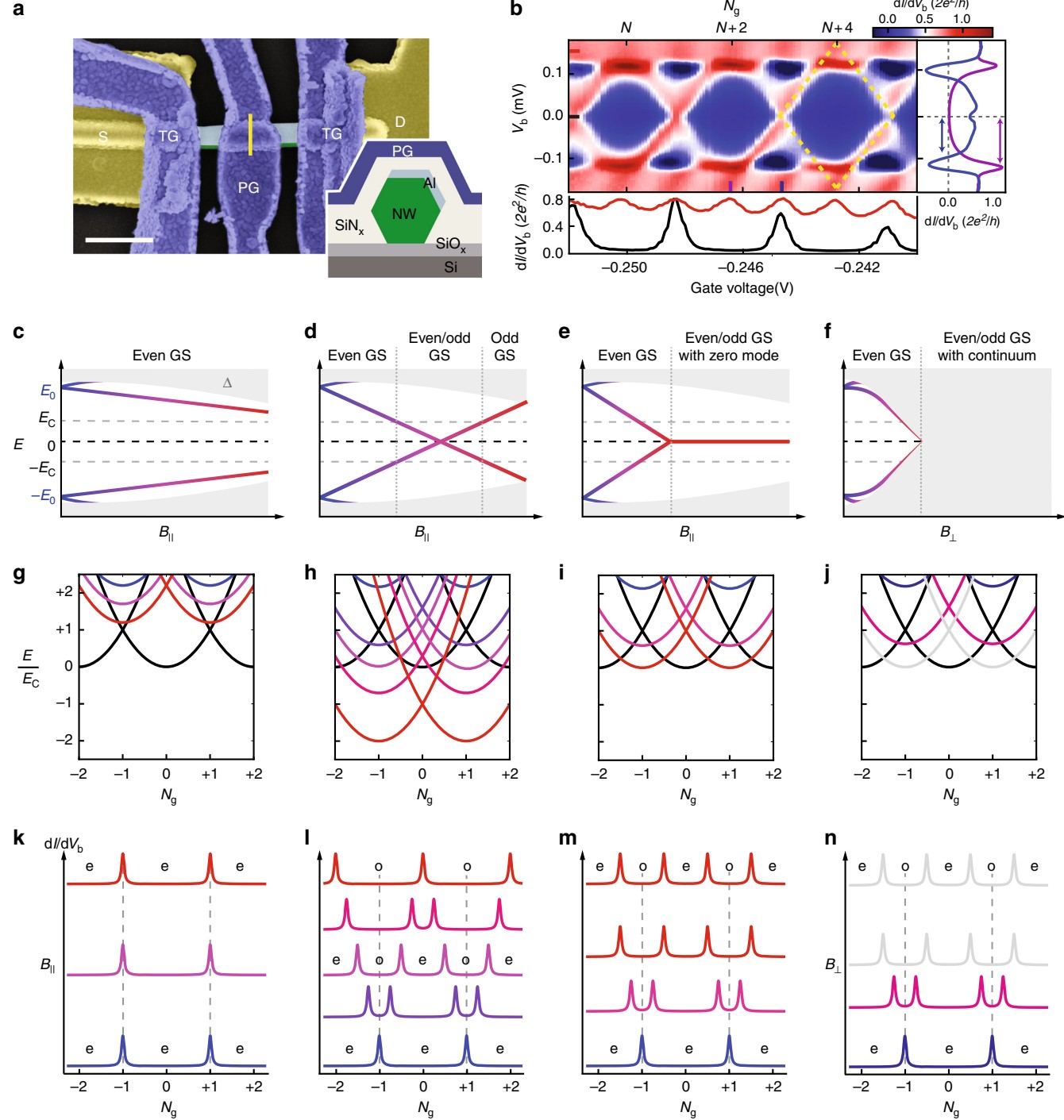

2e-periodic oscillations, but strikingly with an odd-parity in the valleys (cf. Fig. 1l). The parity transition is also illustrated in Fig. 2f by the single crossing between $S_e$ and $S_o$. Similar to Fig. 2a, above 0.9 T the oscillation becomes 1e-periodic (see linecuts).

For the case shown in Fig. 2c (cf. Fig. 1m), the 2e-periodicity gradually changes to uniform 1e-periodicity above ~0.35 T. $S_e$ and $S_o$ exhibit slight but visible parity-changing oscillations up to 0.9 T, whose amplitude decreases with field. This case resembles the experiment of ref. [15], where the 1e oscillations are associated with MZMs. Additionally, we found that the peaks are alternating in height. To quantify this effect, we extract from the data an asymmetry parameter $\Lambda = \frac{G_{e \to o}}{G_{e \to o} + G_{o \to e}}$, which amounts to 0.5 for

peaks with equal heights[22]. Here, $G_{e \to o}$ ($G_{o \to e}$) is the peak height at an even-to-odd (odd-to-even) transition occurring upon increasing $V_{PG}$. Fig. 2g shows that $\Lambda$ undergoes drastic oscillations around 0.5 as $B_{\parallel}$ is varied, and levels off at 0.5 above 0.9 T.

The data in Fig. 2d are taken for the same gate configuration as Fig. 2c but in a perpendicular $B$-field, which turns the Al into a normal state around $B_{\perp} = 0.18$ T (see also Supplementary Fig. 3). In the normal state the oscillations are 1e-periodic and both $\Lambda$ and $S_e/S_o$ are constant, in agreement with established expectations[6].

The four columns in Figs. 1 and 2 represent distinct phases at high $B$: an even-parity GS, an odd-parity GS, a superconducting phase of alternating even and odd parities due to a single state at

**Fig. 1** Hybrid semiconducting–superconducting island and its energy spectrum. **a** False-colour scanning electron microscope image of the device consisting of an InSb nanowire (green) with an 800–900 nm long Al-shell (light-blue) covering the top facet and one side facet. Inset: schematic cross-section at the centre of the plunger gate (PG) indicated by the yellow line. The Si/SiO$_x$ substrate contains a global back gate that we keep at zero voltage. The InSb wire is contacted by Cr/Au leads (yellow) and then covered by a 30 nm-thick dielectric layer of SiN$_x$ (light-grey). Ti/Au top gates (blue) that wrap around the wire allow for local electrostatic control of the electron density. Two gates are used to induce tunnel barriers (TG) and one plunger gate (PG) controls the electron number on the island. The scale bar indicates 500 nm. **b** d$I$/d$V_b$ vs. tunnel gate voltage and $V_b$ showing 2$e$-periodic Coulomb diamonds (one diamond is outlined by yellow dashed lines). The lower panel shows horizontal linecuts with 2$e$-periodic Coulomb oscillations at $V_b = 0$ (black trace) and 1$e$-periodic oscillations at $V_b = 150$ μV (red trace). The panel on the right shows a vertical linecut through the Coulomb peak at the degeneracy point (blue trace) and through the centre of the Coulomb diamond (purple trace). Below are four scenarios for the $B$-dependence of a single Andreev level (**c–f**), the resulting energies as a function of the induced charge, $N_g$ (**g–j**), and the Coulomb oscillations (**k–n**). In panels **c–f**, the grey regions represent the continuum of states above $\Delta$. The coloured traces represent the energy, $E_O$, of the lowest-energy subgap state. Panels **g–j** show the energies of the island with $N$ excess electrons, $E(N_g) = E_c(N_g - N)^2 + p_N E_O$, where $N_g$ is the gate-induced charge, $N$ is the electron occupancy number, and $p_N = 0$ (1) for $N = $ even (odd). Parabolas for $N = $ even are shown in black, while parabolas for $N = $ odd are shown in colours in correspondence to the colours in the other rows. Crossings in the lowest-energy parabolas correspond to Coulomb peaks as sketched in panels **k–n**, again with the same colour coding. Labels in the Coulomb valleys between the peaks indicate the GS parity being either even (e) or odd (o)

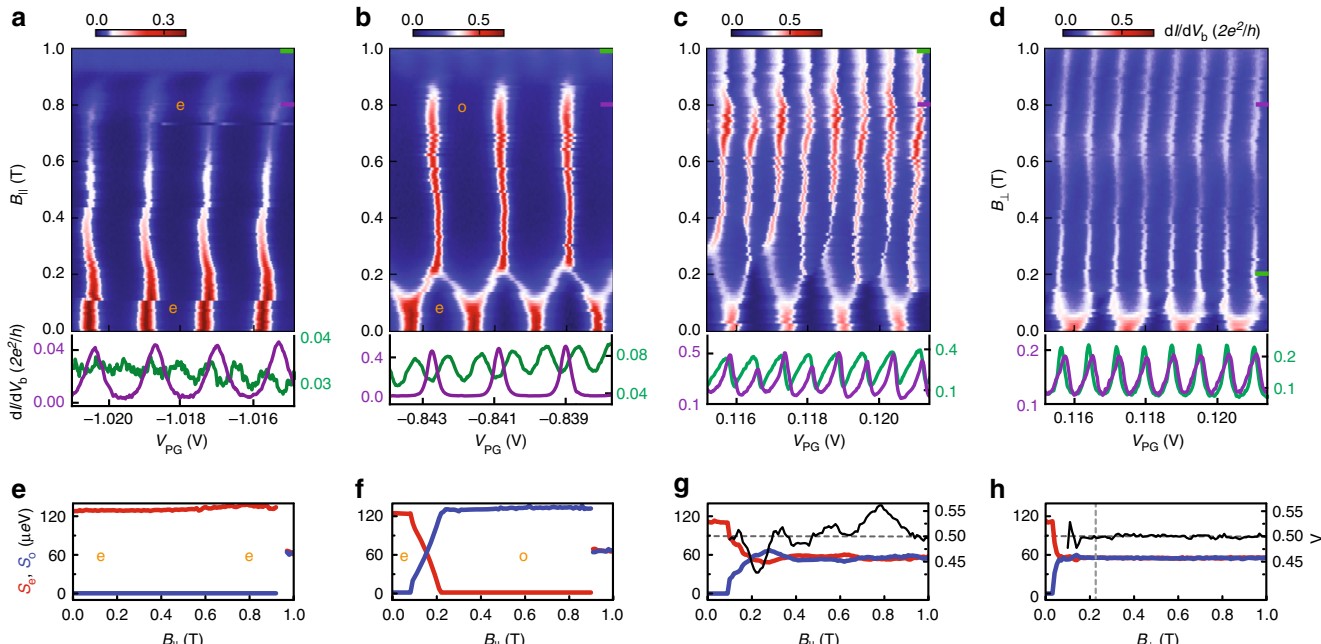

**Fig. 2** Four representative evolutions of Coulomb peaks, corresponding to the four columns (**c–n**) in Fig. 1. Top row panels: d$I$/d$V_b$ as a function of $V_{PG}$ and $B_\parallel$ (**a–c**) or $B_\perp$ (**d**). Below are typical linecuts at different $B$-fields indicated by the purple and green lines. (**e–h**) Even and odd peak spacings, $S_e$ (red) and $S_o$ (blue) on the left axis, and peak height ratio, $\Lambda$ (black) on the right axis, vs. $B$-field, for the valleys labelled e/o in **a** and **b** and for the average spacings in **c** and **d**, respectively. Here and in other figures, the linewidths of $S_e$ and $S_o$ curves correspond to 5 μeV, in accordance with the lock-in excitation energy. **a** The 2$e$-periodicity with even-parity valleys persists up to 0.9 T, above which quasiparticle poisoning occurs. **b** The 2$e$-periodic peaks split at ~0.11 T and merge again at ~0.23 T. For $B_\parallel > 0.23$ T, the oscillations are again 2$e$-periodic, but here the GS parity is odd, consistent with Fig. 1h. **c** 2$e$-periodicity transitioning to uniform 1$e$-periodicity at $B_\parallel \approx 0.35$ T, accompanied by peak spacing and peak height ratio oscillations up to 0.9 T (see also panel **g**). **d** 2$e$-periodicity transitioning to 1$e$-periodicity at $B_\perp = 0.18$ T (the vertical dashed line), coinciding with the critical $B$-field of the Al layer (see Supplementary Fig. 3). Above the critical field, peak heights are constant (see linecuts) and the even/odd peak spacings are equal (**h**). A few common offsets in $V_{PG}$ are introduced to compensate the shifts in gate voltage, and the raw data are listed in Supplementary Fig. 4

zero energy, and a gapless normal phase of alternating even and odd parities. These distinct phases can be reached by varying $V_{PG}$ and thereby the spatial profile of the wave functions, which determines the coupling strength to Al and to the external magnetic field[23–25]. In Fig. 2a, a very negative $V_{PG}$ pushes the wave functions against and partly into the Al, leading to a robust induced gap with weak sensitivity to the $B$-field, which indeed never induces a parity change. Fig. 2b, at more positive $V_{PG}$, reflects the presence of a subgap state with larger weight in the InSb (as indicated by its estimated $g$-factor, $g \approx 7–15$; see another example in Fig. 3f). In Fig. 2c, at even more positive $V_{PG}$,

the involved wave function has an even larger weight in the semiconductor, yielding a large $g$-factor of ~10, possibly augmented by orbital effects[26]. This leads to the appearance of robust zero-energy modes at $B$-fields much lower than the critical field of the thin Al shell. This $V_{PG}$-dependent 2$e$-periodic to 1$e$-periodic transition has been repeated for another device, which is not depicted here.

**2$e$-periodic odd-parity GS.** Figure 3 is dedicated to the new observation of a 2$e$-periodic odd-parity GS. The combined system of a superconducting island weakly coupled to a quantum dot

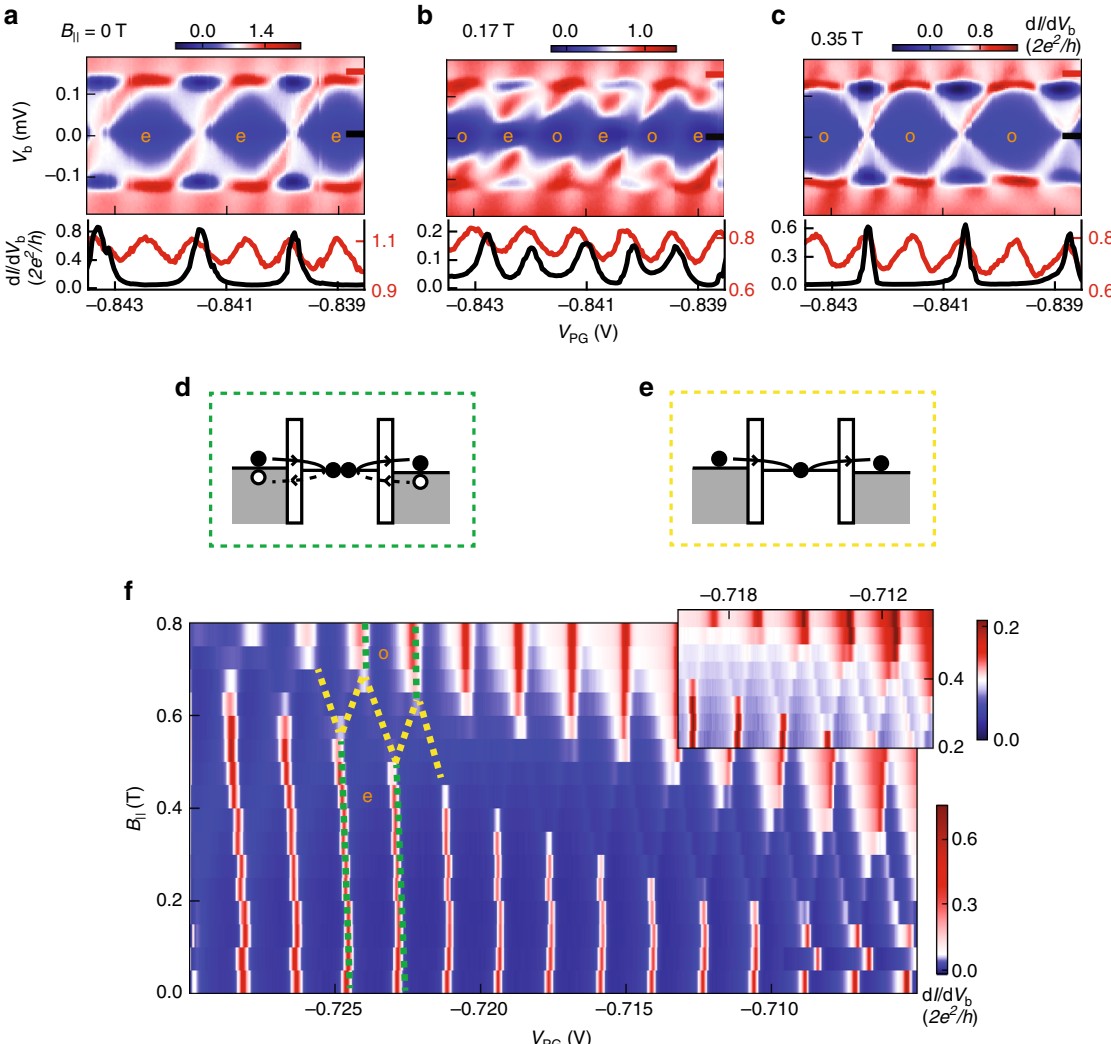

**Fig. 3** Transport via an odd-parity GS. (**a**–**c**) Coulomb diamonds for the gate settings in Fig. 2b at different $B_\parallel$. Below are linecuts at $V_b = 0$ (black trace) and $V_b = 150\,\mu V$ (red trace), respectively. **f** Another example of an even-to-odd GS transition. The inset is a zoom-in of the even–odd regime with a different scale bar. The sketch in **d** illustrates the Cooper pair tunnelling process in both even- and odd-GS regimes (marked by green dashed lines in **f**), while **e** illustrates the single electron tunnelling process in the alternating even–odd parity GS regime (marked by yellow dashed lines in **f**)

with an odd electron number can also have odd parity[27]. In our case all states are strongly hybridised with the superconductor and one bound state drops below $-E_c$, which causes the even-parity state to become an excited state above the odd-parity GS. (We describe different types of bound states in the Supplementary Note 5.) Fig. 3a–c show Coulomb diamonds for the gate settings of Fig. 2b at three values of $B_\parallel$. Fig. 3a at $B_\parallel = 0$ shows 2e-periodic diamonds with 1e and 2e linecuts shown at the bottom (similar data was presented in Fig. 1b). Fig. 3b at $B_\parallel = 0.17$ T corresponds to the even–odd regime. Note that the conductance near $V_b = 0$ is suppressed, indicating that the subgap state causing the even-to-odd transition is weakly coupled to at least one of the normal leads (see also linecuts). Fig. 3c at $B_\parallel = 0.35$ T shows again 2e-periodic diamonds but now the GS inside the diamonds has an odd parity. The diamond structure, including the presence of regions with negative $dI/dV_b$, is very similar to the even-parity GS diamonds, except for the shift in gate charge by 1e.

Figure 3f shows another example of the transition from the 2e-periodic even GS, via a region of even–odd spacings, to a 2e-periodic odd GS. Note again that the even–odd peak heights are significantly suppressed. These peaks correspond to a crossing of an even-parity parabola with an odd-parity parabola in Fig. 1h,

where transport occurs via single electron tunnelling. In contrast, transport at the 2e-periodic peaks, both for even and odd GSs, occurs via Andreev reflection. The two cartoons (Fig. 3d, e) illustrate these different transport mechanisms.

**Isolated zero-energy modes and Coulomb valley oscillations.** Richer sequences of GS transitions as a function of magnetic field are also possible. For instance, the sketch in Fig. 4a illustrates the occurrence of multiple zero-energy crossings at low $B_\parallel$ followed by the appearance of a stable zero-energy state at higher $B_\parallel$. This type of behaviour is observed in Fig. 4b, showing large oscillations of $S_e$ and $S_o$ for $B_\parallel < 0.6$ T, and a stable 1e-periodicity for $B_\parallel > 0.7$ T (see Fig. 4c). Similarly, Fig. 4d also shows large oscillations of $S_e$ and $S_o$ below 0.6 T, followed by a region of almost equally spaced peaks above 0.6 T (see Fig. 4e).

The features at low $B_\parallel$ (such as the position where the 2e-periodic peaks first split) can depend on the precise value of $V_{PG}$. In contrast, the features at high $B_\parallel$ are strikingly regular, with only a weak dependence on $V_{PG}$. This suggests that the 1e-periodicity at high fields originates from a state which is remarkably robust against gate variations. Furthermore, we note

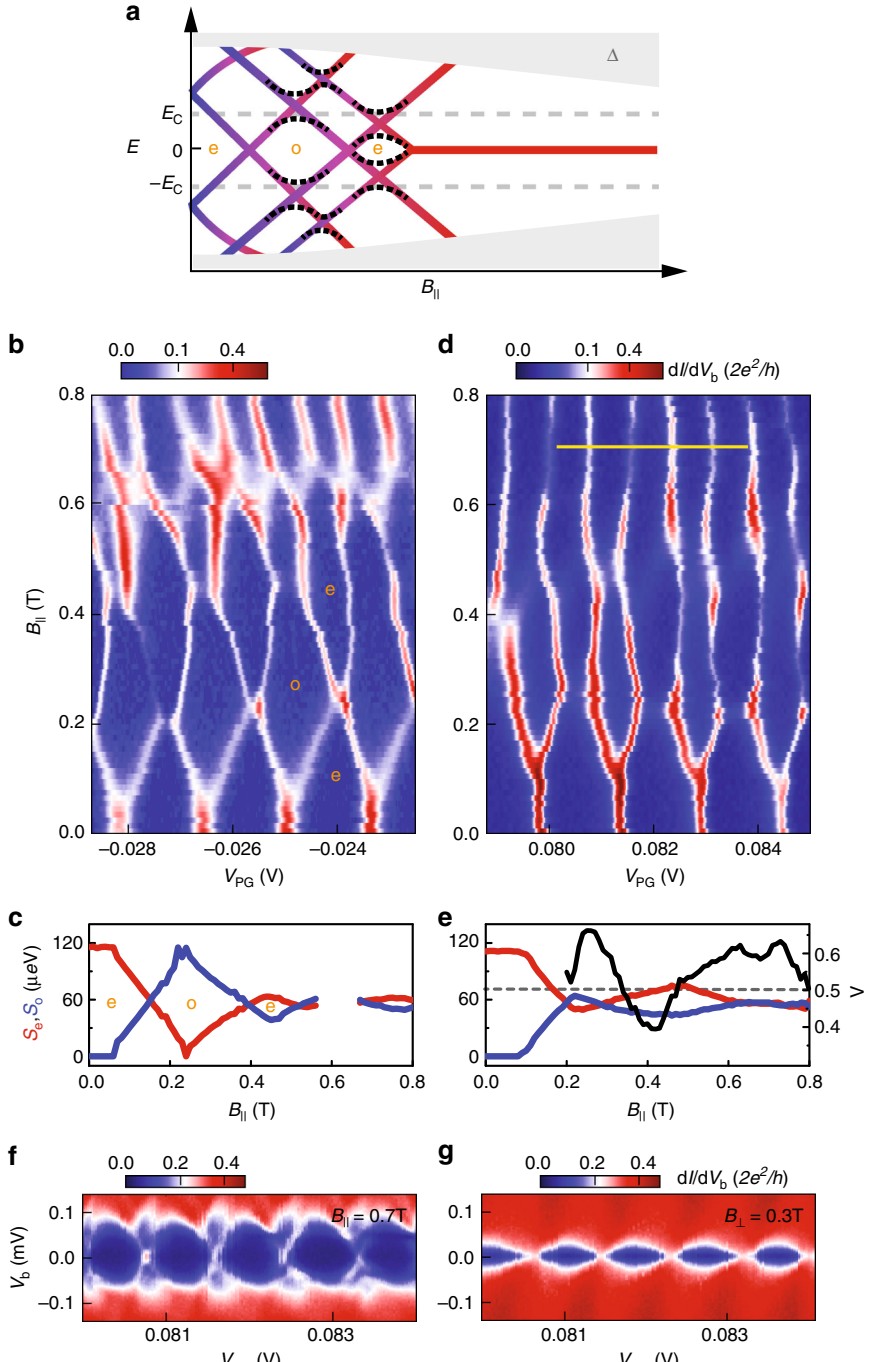

**Fig. 4** Evolution of multiple subgap states. **a** Schematic *B*-field dependence for the case of three subgap states with GS parity transitions at each zero-energy crossing. Dashed lines indicate level repulsion between different subgap states, leading to large oscillations of the lowest energy $E_O$. **b** One example of Coulomb peaks reflected by the scenario in panel **a**. The extracted peak spacings for the valleys labelled by e and o are shown below in **c**. The evolution of the peak spacings is compatible with the type of energy spectrum shown in **a** characterised by large oscillations of $E_O$. Note that the odd-parity GS around ∼0.2 T develops a full 2*e*-periodicity. (The spacings near ∼0.6 T are absent because the exact peak positions are unclear.) **d** Another example of Coulomb oscillations with a pronounced *B*-field dependence. Extracted peak spacings and the height ratio are shown in **e** (averaged over the three periods in **d**). **f** Coulomb diamonds at $B_∥ = 0.7$ T along the $V_{PG}$-range indicated by the yellow line in **d**. This bias spectroscopy reveals isolated zero-bias peaks at the charge-degeneracy points that are separated from the continuum. As in Fig. 2c, neighbouring zero-bias peaks have different heights (also visible in **d** at high $B_∥$). **g** Bias spectroscopy with Al in the normal state ($B_⊥ = 0.3$ T) where the isolated zero-bias peaks are absent

how the alternation of the conductance peak heights, already seen in Fig. 2c, is clearly visible in Fig. 4e even in the 1*e*-periodic regime. The origin of these peak height oscillations lies in the difference between tunnelling amplitudes involving the electron and hole components of the subgap states[22]. It was recently

proposed that in an idealised model for MZMs in a finite-length wire, the oscillations of Λ should be correlated with the oscillations in $S_e$ and $S_o$, i.e. that Λ would be maximal or minimal when $S_e = S_o$ and vice versa that $|S_e − S_o|$ would be maximal for Λ = 0.5[22]. In Fig. 4e, we find that the oscillations in

$\Lambda$ are similar in number and period to the corresponding oscillations in $S_e$ and $S_o$, indeed suggesting a possible connection between the two (another example is presented in Supplementary Figs. 8c–f). Fig. 4f shows finite-bias spectroscopy in the 1$e$-periodic regime of Fig. 4d, at a high parallel field $B_\parallel = 0.7$ T. This spectroscopy reveals that the marked asymmetry of the peak heights originates from a discrete state that is gapped from a continuum of states at higher bias. As a comparison, Fig. 4g shows that for Al in the normal state no discrete features are observed. These are important verifications that substantiate our conclusion that the scenario in Fig. 1e is the proper description of the 1$e$ oscillations in the experimental figures (Figs. 2c and 4d) at high $B_\parallel$.

## Discussion

In summary, we have revealed distinct types of fermion parity transitions occurring as a function of magnetic field and gate voltages in a Coulomb-blockaded InSb–Al island. These transitions provide a complete picture of all the parity phases in mesoscopic Cooper pair boxes[28]. Among these, in a finite field, we find a novel odd-parity phase with 2$e$ periodicity in gate voltage. Additionally, we find 1$e$-periodic oscillations at a high field, which we show to originate from isolated zero-energy modes. The thorough understanding of the involved physics is important, since such islands form the building blocks of future Majorana qubits[29–31].

## Methods

**Device fabrication.** An isolated Al segment is formed by selectively shadowing the nanowire during Al evaporation. InSb–Al nanowires with double shadows (Supplementary Fig. 1) were transferred from the InP growth chip to a doped-Si/SiO$_x$ substrate using a mechanical nanomanipulator installed inside an SEM. Au is used as leads and the top gates. A 30 nm dielectric of SiN$_x$ separates the nanowire from the top gates.

**Transport measurements.** The device is cooled down to ~15 mK in an Oxford dry dilution refrigerator. The effective electron temperature is estimated to be 20–50 mK. Conductance across the devices was measured using a standard low-frequency lock-in technique (amplitude is 5 µV). The voltage bias $V_b = V_S - V_D$ is applied symmetrically between the two leads ($V_S = -V_D = V_b/2$). A magnetic field is applied using a 6-1-1 T vector magnet. The direction of the magnetic field is aligned carefully with respect to the nanowire axis (Supplementary Fig. 3). The sweeping rate of the datapoints is very slow and one $V_{PG}$-dependent linetrace at fixed $B$ in Fig. 2 takes a few minutes. We measure from 0 to 1 T with 0.01 T steps, bringing the measurement time for each panel to 3–4 h. Because of this, we suffer from an ultra-slow drift that effectively changes the island potential. It is important to stress, however, that the slow drift affects the peak positions but not the peak spacings (since each trace only takes a few minutes to acquire). We extract subgap states and parity transitions from peak spacings and thus our conclusions are not affected by the ultra-slow drift.

## Data availability

The figures are created from the raw data. All data and the code used for peak fitting are available at https://doi.org/10.4121/uuid:e0ecafef-7f45-4475-b2f8-351c2af4a2b0.

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

## Acknowledgements

We gratefully acknowledge Joon Sue Lee, Daniel J. Pennachio, and Borzoyeh Shojaei for help with superconductor/semiconductor fabrication and structural characterisation, Petrus J. van Veldhoven for help with nanowire growth, and Kevin van Hoogdalem, Leonid Glazman, and Roman Lutchyn for discussion of the data. This work has been supported by the European Research Council (ERC), the Dutch Organization for Scientific Research (NWO), the Office of Naval Research (ONR), the Laboratory for Physical Sciences and Microsoft Corporation Station Q.

## Author contributions

J.S. fabricated the devices. J.S., F.B., and S.J.J.R. performed the measurements. J.S., S.H., B.V.H. and L.P.K. analysed the data. G.W., D.X., D.B. and A.G. contributed to the discussion of data and the optimisation of the fabrication recipe. S.G., R.L.M.O.H.V.,

D.C., J.A.L., M.P., C.J.P. and E.P.A.M.B. carried out the growth of materials. J.S., B.V.H., S.H., F.B. and L.P.K. co-wrote the paper. All authors commented on the manuscript.

## Additional information

**Competing interests:** The authors declare no competing interests.

