## [Peer Review file · Nature Communications]

Reviewers' comments:

Reviewer #1 (Remarks to the Author):

Dear Editor,

The manuscript titled "Parity transitions in the superconducting ground state of 2 hybrid InSb-Al Coulomb islands" studies the ground state parities of a hybrid superconducting quantum dot as a function of magnetic field. The data shows all three types of Coulomb blockade oscillations i.e. $2e$ periodic between even states, nearly e periodic and alternating in parity and finally the novel $2e$ periodic oscillations between odd states at large magnetic field. Despite the data being superficially similar to Ref [18] (apart from odd parity), the present manuscript provides a clear picture of the level structure of the wire as a function of field. Additionally, I think the $2e$ periodic oscillations in the odd parity sector is definitive evidence for a qualitative effect of a finite magnetic field with the ability to possibly better rule out impurity induced sub-gap states in the device compared to conductance.

In my comments on a previous version of the manuscript, I was worried about certain technical aspects of the data as well as whether the results were significant enough for the broad readership of Nature Physics.

With the revisions, the authors have clarified the issues both referees had raised with the manuscript. The robust odd parity state established clearly in the revised manuscript is an important novel step in the quest of Majorana devices that I can recommend for publication in Nature Communication.

If properly established this could be a way to characterize the energy of the lowest energy spectrum of the quantum dot in a more reliable way than conductance. The manuscript is well-written and clear. However, there are certain aspects of the presentation that leave me unconvinced of the direct relevance of such data to Majorana qubits claimed in the summary. I elaborate on my concerns regarding presentation below.

My main concern is that I am not entirely convinced of the broad significance of the observation $2e$ oscillations in the odd parity sector. As mentioned in the supplementary material, related oscillations have been seen in Ref S5, where of course the structure of the Coulomb interaction is different. The present Coulomb blockade configuration, which is more relevant for the Majorana qubit has already been demonstrated in Ref [18]. So it is a bit unclear if the odd parity oscillations should be considered a major milestone or moderate advance. From the point of view of Majorana qubit devices I would be happier to consider this a major advance if the authors made it clear that the observation of $2e$ periodic oscillations in the odd parity sector is relatively immune to poisoning (along the lines of Ref [19]). Specifically, it appears that the results have been obtained by sweeping V_{PG} (it would be helpful if this were stated explicitly) - the stability of the results to slow sweep speeds would put a bound on the time-scale of switching to e oscillations. In a magnetic field, one expects imperfections somewhere in the device to lead to low energy bound states. These bound

states would be difficult to detect in transport experiment but would destroy the $2e$ periodicity and change it to e periodicity. If the results in the manuscript hold up for slow enough V_{PG} , it would provide a limit on the presence of such states over a range of field. However, since charging energy $E_c \ll \Delta$, this transition happens at a field smaller than where the topological phase might be expected to occur. Emphasizing that such $2e$ periodicity persists for magnetic fields way beyond this transition is (as suggested by Fig 2) would be quite reassuring that the device seems to be immune from at least the component of poisoning that is generated by low energy states.

As mentioned in the previous paragraph - some more details on how V_{PG} is changed would help understand the data better. For example there are clear effects of the shift of V_{PG} from gate charges already mentioned in the paper. This is apparent from the discontinuities in the data. However, many of the conductance plots - such as Fig 2c seem to be shifting to one direction in V_{PG} as $B_{||}$ is changed. This doesn't make sense from the schematic presented in Fig 1. The peaks of $2e$ periodic oscillations peaks should appear at the same value of V_{PG} if the parity doesn't change and the $B_{||}$ dependence of the energy spacing should not be relevant. But clearly the peaks seem to shift with $B_{||}$, which is only consistent with Fig 1 if there is a gate-charge induced shift of the gate potential. It would be useful if the manuscript discusses this.

I think one value of the data in the manuscript is as a characterization of the lowest energy state as a function of gate voltage and magnetic field. So it would be useful if the manuscript commented on the gate voltage dependence. The manuscript is actually very confusing in this regard because the 4 panels in Fig. 2 are just listed as four different examples of the behavior of energy levels in the nanowire. Are these different devices? or are these the same device at different V_{PG} ? If it is the latter then how do the different panels transition to each other?

In line 148, the authors make a deal about the asymmetry parameter Λ . While I understand the operational definition, the manuscript does not provide any insight as to why we care about this parameter. In some weak tunneling limit this parameter might possibly indicate something about the overlap of the states with the tunnel lead the data does not appear to be in such a weak tunneling limit. This is because the strength of the $2e$ periodic oscillations (Fig 2ab) where transport presumably appears through Cooper pair tunneling seems to have the same amplitude as Fig 2c. Additionally, the line-shape of Fig 2c is quite non-Breit-Wigner like.

In fact in Fig. 3b the $2e$ oscillations are stronger than the $1e$ oscillations coming from single electron tunneling. For these reasons it would have been helpful if the authors could provide some information on the tunnel gate dependence of the plots.

In summary, I think the present manuscript represents high quality data which could lead to a detailed understanding of the device. Unfortunately, I don't see the main novelty focus of $2e$ oscillations in an odd parity sector as a significant milestone beyond previous experiments in either the search for Majoranas or as a new phenomenon with deep ramifications. Because of this I feel that the manuscript needs to be revised to provide more insight into the data that might ultimately lead to a better understanding of the level structure of the wires.

Response to the Reviewers' comments

All the response are highlighted in blue fonts

Reviewers Comments:

Reviewer #1 (Remarks to the Author):

Dear Editor,

The manuscript titled "Parity transitions in the superconducting ground state of 2 hybrid InSb-Al Coulomb islands" studies the ground state parities of a hybrid superconducting quantum dot as a function of magnetic field. The data shows all three types of Coulomb blockade oscillations i.e. $2e$ periodic between even states, nearly e periodic and alternating in parity and finally the novel $2e$ periodic oscillations between odd states at large magnetic field. Despite the data being superficially similar to Ref [18] (apart from odd parity), the present manuscript provides a clear picture of the level structure of the wire as a function of field. Additionally, I think the $2e$ periodic oscillations in the odd parity sector is definitive evidence for a qualitative effect of a finite magnetic field with the ability to possibly better rule out impurity induced sub-gap states in the device compared to conductance. If properly established this could be a way to characterize the energy of the lowest energy spectrum of the quantum dot in a more reliable way than conductance. The manuscript is well-written and clear. However, there are certain aspects of the presentation that leave me unconvinced of the direct relevance of such data to Majorana qubits claimed in the summary. I elaborate on my concerns regarding presentation below.

My main concern is that I am not entirely convinced of the broad significance of the observation $2e$ oscillations in the odd parity sector. As mentioned in the supplementary material, related oscillations have been seen in Ref S5, where of course the structure of the Coulomb interaction is different. The present Coulomb blockade configuration, which is more relevant for the Majorana qubit has already been demonstrated in Ref [18]. So it is a bit unclear if the odd parity oscillations should be considered a major milestone or moderate advance. From the point of view of Majorana qubit devices I would be happier to consider this a major advance if the authors made it clear that the observation of $2e$ periodic oscillations in the odd parity sector is relatively immune to poisoning (along the lines of Ref [19]).

Response: We thank the reviewer for pointing this out. There is clear evidence that our $2e$ periodic oscillations in the odd parity sector are indeed immune to poisoning, as demonstrated by the absence in our data of "shadow" features, which are identified with a poisoned state in Ref. 19 and previous literature. This consideration applies both to the zero-bias scans of Fig. 2b and 3b (compare to Fig. 3 of Ref. 19) and to the Coulomb diamond measurements of Fig. 3a (compare to Fig. 2 of Ref. 19). In our experiment poisoning becomes important at B-fields above 0.9 T (Fig. 2a,b) where shadow resonances do become visible. For fields lower than 0.9 T current from quasi-particle poisoning is smaller than our resolution. We thus conclude that the odd-parity $2e$ oscillations are immune to poisoning and this is a major advance for the Majorana qubit devices.

Specifically, it appears that the results have been obtained by sweeping V_{PG} (it would be helpful if this were stated explicitly) - the stability of the results to slow sweep speeds would put a bound on the time-scale of switching to e oscillations.

Response: The V_{PG} sweep rate is very slow and one curve at fixed B in Fig. 2 takes a few minutes. The entire figure of V_{PG} vs B , e.g. Fig. 2b, takes 3-4 hrs. We emphasize that the data is reproducible. We now added the measurement time in the 'Methods' section.

In a magnetic field, one expects imperfections somewhere in the device to lead to low energy bound states. These bound states would be difficult to detect in transport experiment but would destroy the $2e$ periodicity and change it to e periodicity. If the results in the manuscript hold up for slow enough V_{PG} , it would provide a limit on the presence of such states over a range of field. However, since charging energy $E_c \ll \Delta$, this transition happens at a field smaller than where the topological phase might be expected to occur. Emphasizing that such $2e$ periodicity persists for magnetic fields way beyond this transition is (as suggested by Fig 2) would be quite reassuring that the device seems to be immune from at least the component of poisoning that is generated by low energy states.

Response: As we replied above the results indeed hold for very slow V_{PG} sweep rates and we agree with the referee that the $2e$ results in Fig. 2 reassure that our device is immune from poisoning due to low energy states.

As mentioned in the previous paragraph - some more details on how V_{PG} is changed would help understand the data better. For example there are clear effects of the shift of V_{PG} from gate charges already mentioned in the paper. This is apparent from the discontinuities in the data. However, many of the conductance plots - such as Fig 2c seem to be shifting to one direction in V_{PG} as $B_{||}$ is changed. This doesn't make sense from the schematic presented in Fig 1. The peaks of $2e$ periodic oscillations peaks should appear at the same value of V_{PG} if the parity doesn't change and the $B_{||}$ dependence of the energy spacing should not be relevant. But clearly the peaks seem to shift with $B_{||}$, which is only consistent with Fig 1 if there is a gate-charge induced shift of the gate potential. It would be useful if the manuscript discusses this.

Response: Since these 2D data sets take several hours to acquire we suffer from some ultra-slow drift that effectively changes the island potential. It is important to stress, however, that slow drift affects the peak positions but not the peak spacings (since each trace only takes a few minutes to acquire). We extract subgap states and parity transitions from peak spacings and thus our conclusions are not affected by ultra-slow drift. We now added this in the 'Methods' section.

I think one value of the data in the manuscript is as a characterization of the lowest energy state as a function of gate voltage and magnetic field. So it would be useful if the manuscript commented on

the gate voltage dependence. The manuscript is actually very confusing in this regard because the 4 panels in Fig. 2 are just listed as four different examples of the behavior of energy levels in the nanowire. Are these different devices? or are these the same device at different VPG? If it is the latter then how do the different panels transition to each other?

Response: We thank the Referee for this very important question. All the figures in the main text are from the same device, including the four panels in Fig.2. The panels in Fig. 2 are taken at different gate voltages which allows us to reach different regimes in a single device. Note that the panels a and b in Fig. 2 are separated by ~ 100 Coulomb valleys, and panels b and c are even further apart. The different regimes can be understood to be due to moving the electron wave function from the AI into the nanowire, as discussed in the main text. We note that we have reproduced the entire experiment in two other devices. We added the sentence ‘This V_{PG} – dependent $2e$ - to $1e$ -periodic transition has been repeated in two devices.’ in paragraph 14.

In line 148, the authors make a deal about the asymmetry parameter Λ . While I understand the operational definition, the manuscript does not provide any insight as to why we care about this parameter.

Response: We agree with the referee that the consequences of the Λ parameter are somewhat unclear. We have included this nevertheless since it was recently predicted (based on a simplified model) that this parameter could be an indicator for Majorana states. Our data allows to determine Λ and we feel obliged, in light of these predictions, to include it in our manuscript. The model in [28] however seems to be too simplified to capture our experiment, but still we feel we should include this ‘negative’ result.

In some weak tunneling limit this parameter might possibly indicate something about the overlap of the states with the tunnel lead the data does not appear to be in such a weak tunneling limit. This is because the strength of the $2e$ periodic oscillations (Fig 2ab) where transport presumably appears through Cooper pair tunneling seems to have the same amplitude as Fig 2c. Additionally, the line-shape of Fig 2c is quite non-Breit-Wigner like. In fact in Fig. 3b the $2e$ oscillations are stronger than the $1e$ oscillations coming from single electron tunneling. For these reasons it would have been helpful if the authors could provide some information on the tunnel gate dependence of the plots.

Response: The relative height of conductance peaks in the different transport regimes of hybrid nanowire islands is an open and interesting question. We find that a systematic investigation of this topic is made difficult by the fact that upon varying the tunnel gates we observe fluctuations of the average conductance level of the device. This behavior is illustrated in the measurement shown below. The gate and field dependence of the tunnel couplings is non-monotonic, possibly due to the appearances of resonances in the tunnel barriers. This non-systematic behavior makes it hard to compare peak height or shape at different voltages or magnetic fields, and so we have decided not to discuss this aspect of the data in the manuscript.

In summary, I think the present manuscript represents high quality data which could lead to a detailed understanding of the device. Unfortunately, I don't see the main novelty focus of $2e$ oscillations in an odd parity sector as a significant milestone beyond previous experiments in either the search for Majoranas or as a new phenomenon with deep ramifications. Because of this I feel that the manuscript needs to be revised to provide more insight into the data that might ultimately lead to a better understanding of the level structure of the wires.

Reviewer #2 (Remarks to the Author):

Title: Parity transitions in the superconducting ground state of hybrid InSb-Al Coulomb islands

Jie Shen et al.

In this work the authors characterize the electrical properties of a charge island (box) formed by a hybrid system that consist of an InSb nanowire (NW) coated in part with an epitaxial thin film of Al which is a superconductor (SC) at low temperatures. Using top gates to the NW they can define two tunneling barriers to this “hybrid charge box”, one at each side. This yields a superconducting “single-electron-transistor” (SET) device. Generically, the gate- and bias-dependent conductance through such a superconducting SET reveals the periodic addition of charge in units of $2e$, which is a very distinct feature relative to an SET in the normal state, where charge is added in units of e .

One first should say here, that the authors redo the same experiments as done by Albrecht et al. (Nature V531, 2016 & Phys. Rev. Lett. 2017). The difference is that the Albrecht measurements were done on InAs-Al islands and the one here on InSb-Al islands. In both cases, the $2e$ addition is confirmed at low enough temperature, weak coupling and zero magnetic field. With increasing magnetic field a transition to a non-trivial $1e$ periodic state is demonstrated in the work of Albrecht et al. and interpreted as due to the emergence of a Majorana bound state, i.e. a quasiparticle state that resides close to zero energy. It has also been shown before that due to the trapping of a

quasiparticle the ground-state (GS) changes parity from even to odd. However, in previous work this transition was dynamical, not really controlled on a “long” time scale. What is new in the present work is that the author presents data where a magnetic-field dependent in-gap state (an Andreev bound state) moves with field through zero energy changing the GS from even to odd (or from odd to even). This is a stable transition that can be well controlled and persist on a “long” time scale. The present story is convincing and the paper is written in a very clear manner. All now boils down to the question whether this new work constitutes a breakthrough worth a publication in Nature Physics?

The author have realized hybrid InSb-Al charge boxes that seem “un-poisoned” on the time scale of their measurements (it would be good in future experiments, if one would report the measurement time).

Response: Thanks for pointing this out. We now added the measurement time-scale in the ‘Methods’ section.

This result is nice, but only in part new. Looking at Fig.1 one could say that out of the four scenarios c) – f), the new observation is d). All others have been seen before, also in hybrid NW charge boxes. Also, the GS transition is not unexpected in a material with a large g-factor. What speaks in favor of this paper, is that the authors are able to spectroscopically study the evolution of ABSs that live under the AI SC. Further, if there are more ABSs, the transition can be complex as shown in Fig.4. I personally think the authors present wonderful data, a nice story, nicely presented, but I don’t think it is a major breakthrough. This paper would fit very well to Phys. Rev. Lett.

Revision in the manuscript:

1. We have added ‘The sweeping rate of datapoints is very slow and one VPG-dependent linetrace at fixed B in Fig. 2 takes a few minutes. We measure from 0 to 1T with 0.01T steps, bringing the measurement time for each panel to 3-4 hours. Because of this, we suffer from some ultra-slow drift that effectively changes the island potential. It is important to stress, however, that slow drift affects the peak positions but not the peak spacings (since each trace only takes a few minutes to acquire). We extract subgap states and parity transitions from peak spacings and thus our conclusions are not affected by ultra-slow drift. So each panel in Fig. 2 takes 3-4 hours.’ in the ‘Methods’ section.
2. We have added the sentence ‘This V_{PG} – dependent $2e^-$ to $1e^-$ -periodic transition has been repeated in two devices.’ in paragraph 14.